# Identification of the Interaction between Minichromosome Maintenance Proteins and the Core Protein of Hepatitis B Virus

**Kaili Du** [1], **Eriko Ohsaki** [1,2,*], **Masami Wada** [1] **and Keiji Ueda** [1,*]

1   Division of Virology, Department of Microbiology and Immunology, Osaka University Graduate School of Medicine, 2-2 Yamada-oka, Suita 565-0871, Japan
2   Center for Infectious Disease Education and Research, Osaka University (CiDER), 2-2 Yamada-oka, Suita 565-0871, Japan
*   Correspondence: eohsaki@virus.med.osaka-u.ac.jp (E.O.); kueda@virus.med.osaka-u.ac.jp (K.U.)

**Abstract:** Chronic HBV infection is a major cause of cirrhosis and hepatocellular carcinoma. Finding host factors involved in the viral life cycle and elucidating their mechanisms is essential for developing innovative strategies for treating HBV. The HBV core protein has pleiotropic roles in HBV replication; thus, finding the interactions between the core protein and host factors is important in clarifying the mechanism of viral infection and proliferation. Recent studies have revealed that core proteins are involved in cccDNA formation, transcriptional regulation, and RNA metabolism, in addition to their primary functions of capsid formation and pgRNA packaging. Here, we report the interaction of the core protein with MCMs, which have an essential role in host DNA replication. The knockdown of MCM2 led to increased viral replication during infection, suggesting that MCM2 serves as a restriction factor for HBV proliferation. This study opens the possibility of elucidating the relationship between core proteins and host factors and their function in viral proliferation.

**Keywords:** hepatitis B virus; HBV; core protein; minichromosome maintenance proteins; MCMs; DNA replication

## 1. Introduction

Approximately 350 million people worldwide have chronic the hepatitis B virus (HBV) infection, a disease that causes acute and chronic necro-inflammatory liver disease because of its organ tropism. Controlling HBV infection is considered an urgent issue by the World Health Organization [1,2]. Currently, thorough measures to prevent HBV infection are still insufficient, as is the establishment of therapeutic methods to eradicate the virus post-infection [3]. It is critical to these efforts that researchers elucidate the details of the HBV infection cycle and its infectious pathology inside host cells.

After infection to susceptible cells, nucleocapsids are released into the cytoplasm, and, then, the relaxed circular (rc) HBV DNA is transported into the nucleus, where it is repaired by cellular DNA repair factors to form a covalently closed circular (ccc) DNA [4]. Viral RNAs are transcribed from the cccDNA to produce virion, including pregenomic RNA (pgRNA), which serves as a template for the viral reverse transcription and as an mRNA for the translation of core and polymerase protein [5]. The well-known primary function of the core protein is to form nucleocapsids, where the viral DNA is replicated by reverse transcription [5,6]. The core proteins compose the HBV capsid, which has icosahedral symmetry, made up of 90 or 120 core protein homodimers [7]. The full-length core protein has 183 amino acids, all of which contain an assembly domain (amino acids 1–149) and a nucleic acid-binding domain (amino acids 150–183) [6].

In addition to its role in nucleocapsid formation, multiple possible roles in cccDNA function and the interaction with cellular proteins have been reported. Bock et al. reported that the HBV core protein binds to the HBV double-stranded DNA and re-arranges the HBV minichromosome by regulating the nucleosomal spacing, and is involved in viral

transcriptional regulation [8]. A recent study reported that the phosphorylation of the assembly domain of the core protein affects cccDNA synthesis [9].

As evidenced by the studies mentioned above and the fact that core protein allosteric modulators (CpAMs) are promising as key components of hepatitis B curative therapies [10], identifying factors that interact with core proteins is an attractive approach for therapeutic drug development. We have previously attempted to identify core protein interaction factors using highly purified core proteins. Mass spectrometry has identified several promising interaction factors, including the minichromosome maintenance proteins (MCMs), which are components of the pre-replicative complex (pre-RC).

Here, we demonstrate the interaction of the HBV core protein with MCMs and the inhibitory effects of MCM2 on HBV proliferation. This is the first report showing that MCMs interact with the HBV core protein. Thus, these results have implications for our understanding of the unknown functions of core proteins in viral replication.

## 2. Materials and Methods

### 2.1. Plasmids

To establish a lentiviral vector to transduce the $6\times$ His core (HisCore) into mammalian cultured cells, we cloned a $6\times$ His-tagged HBV core ORF (HBV subtype adr4) [11,12] into the XhoI-XbaI site of a pLVSIN-CMV Hyg vector (Takara-Clontech, Shiga, Japan) to generate pLVSIN-CMV Hyg HisCore. All the sequences were confirmed by sequencing analysis.

The plasmid pLVSIN CMV Hyg HisLacZ was constructed for use as a control. A His-lacZ fragment was prepared from pEBV His-lacZ (Invitrogen, Waltham, MA, USA) with restriction enzymes, SphI and HindIII. The terminus of the fragment was repaired and cloned into the repaired BamHI site of pLVSIN CMV Hyg (Takara-Clontech, Shiga, Japan).

The terminal Halo-tagged human MCM2-, 3-, 4-, 5-, 6- and 7-expressing plasmids (pFN21AA0030 for MCM2, FN21AB5749 for MCM3, FN21AE2132 for MCM4, FN21AB4809 for MCM5, FN21AE2394 for MCM6, FN21AE2808 for MCM7, respectively) were all purchased from the Kazusa ORFeome Project (Promega, Tokyo, Japan).

To establish the MCM2 knocked down cells in NTCP-expressing HepG2 cells, shRNA-expressing lentiviral vector plasmids were purchased from a manufacturer (Applied Biological Materials Inc., Richmond, BC, Canada). The target sequences were a: 5′-ggatggagaggacctcatt-3′, b: 5′-ctatcagaactaccagcgtat-3′, and c: 5′-gctcttcatactgaagacagtt-3′. A control shRNA (scrambled)-expressing lentiviral vector plasmid was also purchased from a manufacturer (Applied Biological Materials Inc., Richmond, BC, Canada).

### 2.2. Cells

The human hepatocellular carcinoma cell line HepG2 was grown in low-glucose (1.0 g/L) Dulbecco's modified Eagle's medium (DMEM) (Nacalai Tesque, Kyoto, Japan), supplemented with 10% fetal bovine serum (FBS) (Sigma-Aldrich, St. Louis, MO, USA) and an antibiotic-antimycotic mixed solution (10 IU/mL penicillin G, 10 μg/mL of streptomycin, and 0.25 μg/mL amphotericin B) (Nacalai Tesque, Kyoto, Japan). HepG2/HisCore cells (see below) were cultured with the same low-glucose DMEM, except for the addition of 0.5 mg/mL hygromycin (hyg) B (FUJIFILM Wako Pure Chemicals, Osaka, Japan).

HEK293T cells (Takara-Clontech, Shiga, Japan) were grown in DMEM (high glucose; 5.0 g/L) (Nacalai Tesque, Kyoto, Japan), supplemented with 10% FBS (Sigma-Aldrich, St. Louis, MO, USA), and the antibiotics mentioned above. LentiX-293T cells were maintained in DMEM (high glucose; 5.0 g/L) (Nacalai Tesque, Kyoto, Japan), supplemented with 10% FBS (Sigma-Aldrich, St. Louis, MO, USA) and the antibiotic–antimycotic mixed solution mentioned above.

NTCP-HepG2 cells (NTCP/HepG2 or C4) [13] were grown in William's E medium (Gibco, Waltham, MA, USA), supplemented with 10% FBS (Sigma-Aldrich, St. Louis, MO, USA), 2 mM L-glutamine (Nacalai Tesque, Kyoto, Japan), 50 μM hydrocortisone (Sigma-Aldrich, St. Louis, MO, USA), 5 μg/mL insulin (Sigma-Aldrich, St. Louis, MO, USA), 10 ng/mL epidermal growth factor (EGF) (Fisher Scientific, Waltham, MA, USA),

5 µg transferrin (Wako Pure Chemicals, Osaka, Japan), and 5 ng/mL sodium selenite (Sigma-Aldrich, St. Louis, MO, USA) (primary hepatocyte maintenance medium [PMM]).

### 2.3. Transfection

The transient transfection of plasmids of pLVSIN HisCore and Halo™-tagged human MCMs in HEK 293T cells was conducted with TransIT-LT1™ (Mirus Bio LLC, Madison, WI, USA), according to the manufacturer's instruction.

Lentivirus vectors were prepared by transfecting either pLVSIN HisCore or pLVSIN HisLacZ into LentiX-293T cells (Takara-Clontech, Shiga, Japan) using Lenti-X packaging single shots (VSV-G) (Takara-Clontech, Shiga, Japan), according to the manufacturer's instructions. A few days after transfection, the supernatant was collected and filtered with a 0.22-µm filter and used for transduction experiments. Lentiviral vectors, used to express shMCM2, were prepared in the same way.

### 2.4. Establishment of Stably HisCore and HisLacZ Expressing Cells

The HepG2 cells were transduced by lentiviral vectors to express HisCore or HisLacZ. After two days of transduction, the cells were selected with 0.5 mg/mL hyg B-containing culture medium, and polyclones of HepG2 cells expressing either HisCore or HisLacZ were established; these were referred to as HepG2/HisCore or HepG2/HisLacZ. The expression was confirmed with Western blot analysis and immunofluorescent analysis (IFA) (see below).

### 2.5. Establishment of MCM2 Knocked Down NTCP Expressing HepG2 Cells (C4 Cells)

The lentiviruses, used to express shRNA against human MCM2, were infected with C4 cells. After infecting for 2 days, the cells were selected in the PMM containing 0.5 mg/mL G418 and 5 µg/mL puromycin (Takara-Clontech, Shiga, Japan).

### 2.6. CsCl Density Profile of HisCore

HEK293T/HisCore and HepG2/HisCore cells were collected from semi-confluent 10-cm dishes and lysed with the lysis buffer mentioned above. In total, 500-µL cell lysates without debris were layered on a CsCl gradient solution (39F: 500 µL, 35F: 500 µL, 31F: 300 µL, 27F: 200 µL, 23F: 100 µL and 19F: 100 µL from the bottom, where F means $w/w$%). Then, the samples were centrifuged at 50,000 rpm at 15 °C for 20 h. After the ultracentrifugation, the samples were fractionated from the top into 14 fractions. The refraction of each fraction was measured with a refractometer, and the density was calculated according to the equation: ρg/mL at 25 °C = 10.8601 × $\eta$ − 13.4974, here $\eta$ means refraction). Next, 10 µL of each fraction was subjected to HBeAg ELISA. The remaining sample was concentrated and subjected to Western blotting.

### 2.7. Pull-Down Assay

For the pull-down assay of endogenous MCM2 and MCM5, HepG2/HisCore cells were used. The collected cells were lysed using a lysis buffer (10 mM Tris-HCl pH 7.4, 150 mM NaCl, 1% NP-40, 1% glycerol, 0.1% protease inhibitor cocktail for His-tag purification [Sigma-Aldrich, St. Louis, MO, USA]) and applied on Ni-NTA beads overnight at 4 °C; then, the beads were washed 6–8 times with a washing buffer (10 mM Tris-HCl pH 7.4, 150 mM NaCl, 0.5% NP-40, 1% glycerol, 0.1% protease inhibitor cocktail for His-tag purification). Finally, the protein complexes were eluted with 100 µL of 1× sample buffer (diluted from 5× sample buffer [0.225 M Tris-HCl pH 6.8; 50% glycerol; 5% SDS; 0.05% bromophenol blue; 0.25 M DTT, pH 6.8]) and boiled for 10 min at 100 °C.

For the pull-down assay, using human Halo™-MCM2, 5 µg of a human Halo™-MCM2 expression vector was transiently transfected into HEK293T using TransIT LT1™ overnight. The cells (6 × 10⁶ cells on a 10 cm dish) were prepared one day before transfection. Forty-eight hours post-transfection, the cells were harvested and washed with phosphate-buffered saline (PBS) once and lysed in a lysis/binding buffer (1.3 mL) (mentioned above); next,

600 μL of the lysate was co-incubated either with HepG2/HisCore or HepG2/HisLacZ lysate, prepared from approximately $3 \times 10^6$ cells with 60 μL Halo-link resin (Promega, Tokyo, Japan). The samples were incubated with agitation overnight at 4 °C and then they were washed 6–8 times with the washing buffer mentioned above. Finally, the protein complexes were eluted with 85 μL of 1× sample buffer and boiled for 10 min at 100 °C to separate on SDS-PAGE, followed by blotting analysis. An anti-Halo™ antibody (mouse) was used to detect HaloMCM2 and a mouse anti-His polyclonal Ab (MBL) was used to detect His-tag. Specific antibodies against the other MCM (3 to 7) were used for detection.

### 2.8. Western Blot Analysis

Protein samples were separated on SDS-PAGE, then transferred to a polyvinyl difluoride (PVDF) membrane (ImmunoBlot PVDF; Bio-Rad, Hercules, CA, USA). After blocking with a blocking buffer (5% dry milk [Nacalai Tesque, Kyoto, Japan] containing TBS-T [20 mM Tris-HCl pH 7.6, 150 mM NaCl, and 0.1% Tween™ 20]) for 60 to 90 min, the membranes were washed 3 times with TBS-T and incubated with primary antibodies overnight at 4 °C. The first antibodies were washed out; then, secondary anti-mouse or anti-rabbit IgG, conjugated with horseradish peroxidase (HRP) (Dako, Santa Clara, CA, USA), were allowed to react for 2 h at room temperature (RT). After washing the membranes at least 3 times with TBS-T, chemiluminescence was generated by a commercial kit (Clarity Western ECL Substrate; Bio-Rad, Hercules, CA, USA). Luminescence images were obtained with an imager (ChemiDoc, Bio-Rad, Hercules, CA, USA).

### 2.9. Immunofluorescence Assay

Each human Halo™-tagged MCM2/3/4/5/6/7 expression vector was co-transfected into HEK293T cells with the HisCore, used for establishing stable core-expressing cell lines. At 24–48 h post-transfection, the cells were fixed with 4% paraformaldehyde (PFA) in PBS for 30 min at 4 °C, then permeabilized with 0.1% Triton™ X-100 and 2.5% BSA in PBS for 60 min at RT and washed at least 3 times with PBS. The cells were then probed with rabbit anti-core polyclonal antibodies (Beacle, Inc., Kyoto, Japan) and a mouse monoclonal anti-Halo™-tag antibody (Promega, Tokyo, Japan), diluted in PBS containing 0.3% Triton™ X-100 and 1% BSA, and were incubated at 4 °C overnight. The cells were washed at least 3 times with PBS and incubated with goat Alexa™ Fluor 488 conjugated anti-mouse and/or goat Alexa™ Fluor 546 conjugated anti-rabbit antibodies for 2 h at RT. After being washed at least 3 times with PBS, the cells were mounted with DAPI (4′,6-diamidino-2-phenylindole)-containing glycerol solution (Fluoro-KEPPER Antifade Reagent; Nacalai Tesque, Kyoto, Japan). Images were acquired using a laser scanning confocal microscope (Leica TCS SP8, Wetzlar, Germany).

### 2.10. Enzyme-Linked Immunosorbent Assay (ELISA)

The lysate HBeAg was measured with a commercial ELISA kit (HBeAg Diagnostic Kit [RO Bio, Shanghai, China]), according to the manufacturer's instructions. This ELISA also detected the HBV core protein. ELISA data were obtained as optical density (OD) at 450 nm, and 630 nm was used as a reference.

### 2.11. HBV Infection

For the HBV infection experiment, $2 \times 10^5$ cells/well of MCM2-knocked down C4 cells were seeded in a collagen-coated 24-well plate (Iwaki Cytec, Shanghai, China) the day before infection. HBV viral particles prepared from HepAD W38.7 cells at a 1000 genome equivalent of infection (GEI) in 500 μL of the PMM containing 2% dimethyl sulfoxide (DMSO) and 4%PEG 8000. The next day after the HBV infection, the infected cells were washed at least twice with the PMM containing 2% DMSO and replaced with 1 mL of the washing medium. The culture medium was refreshed every 3 days until 9 days post-infection (dpi) with the PMM containing 2% DMSO. At 9 dpi, the culturing supernatants and cells were collected to evaluate levels of HBeAg expression, covalently closed circular

DNA (cccDNA), extracellular particle-associated HBV DNA, and core-associated HBV DNA. During the 9-day course of the infection, the cells were also collected at days −1, 0, 3, 6, and 9 to monitor the knockdown efficiency of MCM2.

### 2.12. Quantitative Real-Time PCR (qPCR)

Core-associated HBV DNA and extracellular HBV DNA were quantified by Quantitative real-time PCR (qPCR), using Fast SYBR Green Master Mix (Applied Biosystems, Waltham, MA, USA) with a set of S region primers, 5′-CTTCATCCTGCTGCTATGCCT-3′ and 5′-AAAGCCCAGGATGATGGGAT-3′, in order to quantify core and extra-cellular particle-associated HBV DNA. In the case of cccDNA quantification, the total DNA was extracted from the infected cells at day 9. After measuring the total DNA concentration, approximately 1 μg of DNA was treated with ATP-dependent plasmid-safe DNase (Plasmid-Safe™ ATP Dependent DNase, Epicenter®); the primers 5′-GTCTGTGCCTTCTCATCTGC-3′ and 5′-GCACAGCTTGGAGGCTTGAA-3′ were used for quantifying cccDNA. The values for cccDNA were normalized and shown per μg of total DNA.

### 2.13. Antibodies

The following antibodies (Ab) were used in this study: a rabbit anti-Core polyclonal Ab (Beacle Inc., Kyoto, Japan), a mouse anti-His polyclonal Ab (MBL), a mouse anti-Halo monoclonal antibody (mAb) (Promega, Tokyo, Japan), a mouse anti-β-tubulin mAb (Sigma-Aldrich, St. Louis, MO, USA), a rabbit anti-MCM2 mAb (Cell Signaling, Danvers, MA, USA), a rabbit anti-MCM3 mAb (Cell Signaling), a rabbit anti-MCM4 mAb (Cell Signaling), a rabbit anti-MCM7 mAb (Cell Signaling), a mouse anti-MCM5 mAb; and a rabbit anti-MCM6 mAb (Abcam, Cambridge, UK).

### 2.14. Data Processing and Analysis

Quantitative variables were shown by mean ± SD, and the Student's *t*-test was used to compare statistical discrepancies among groups, with $p < 0.05$ regarded as statistically significant values.

## 3. Results

### 3.1. Establishment of Human Hepatoma Cell Line Stably Expressing Core Protein

To examine the interaction between the core protein and the MCM protein, we generated stable cell lines in which the core proteins were continuously expressed using human hepatocytes (HepG2). We first confirmed the core protein expression and core particle formation in HEK293T cells with (Figure 1a). The cesium chloride (CsCl) density gradient ultracentrifugation was used for the separation of core particles. As shown in Figure 1a, the HBe antigen peak coincided with the fraction in which core proteins accumulated, whose density was approximately 1.26 g/mL, suggesting that the His-tagged core proteins constituted capsids in the HEK293T cells. After establishing stable cell lines expressing the core protein, the core particle formation in HepG2/HisCore cells (Figure 1b) was also confirmed by CsCl density gradient ultracentrifugation, as in the HEK293T cells. A purification of HisCore or HisLacZ with Ni-NTA was also conducted (Figure 1c). These results verified that the His-tagged core protein was stably expressed in human hepatocytes and formed core particles.

### 3.2. HBV Core Protein Interacts with Endogenous MCM2

To examine whether endogenous MCM2 interacts with HisCore in human hepatocytes, we performed a pull-down assay using HepG2/HisCore cells (Figure 2). As shown in Figure 2, endogenous MCM2 was pulled down along with the HisCore protein (Figure 2, lane 2), but not with the control protein HisLacZ (Figure 2, lane 4). MCM5, which is one of the components of the helicase MCM2-7 complex, was also detected in the bound fraction. From these results, the HBV core protein was confirmed to interact with human MCM2, and with another MCM component, MCM5.

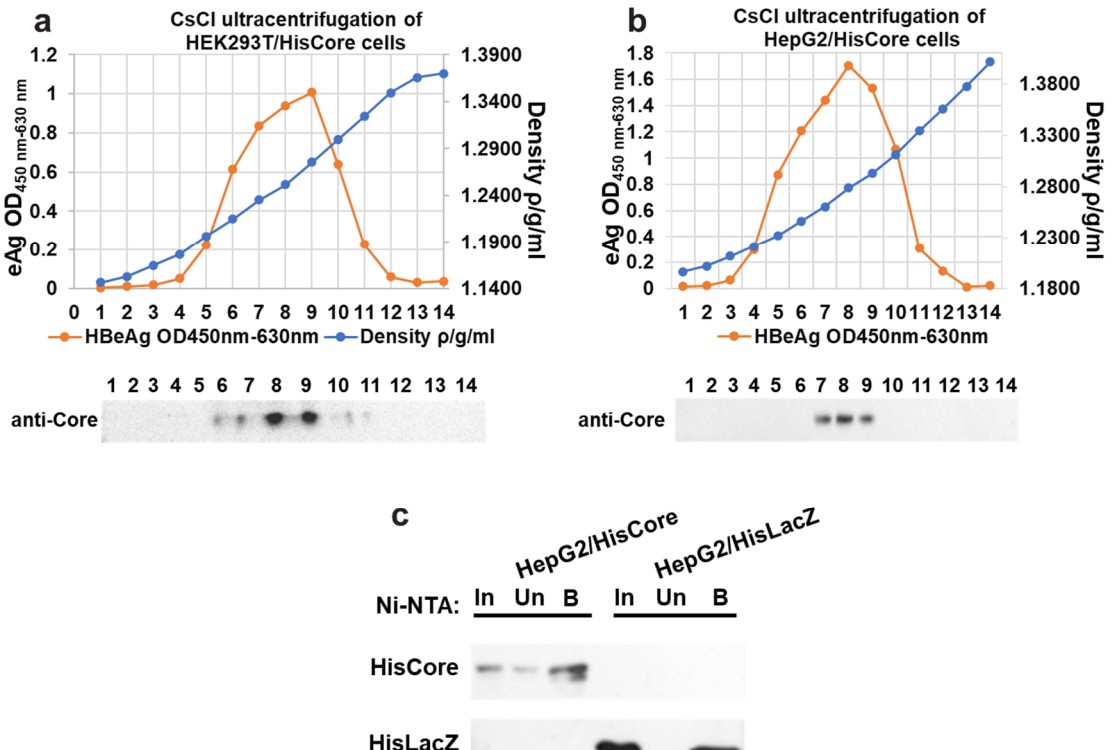

**Figure 1.** Fractionation of core particles by CsCl density gradient ultracentrifugation in transiently transfected HEK293T cells (**a**), and stably expressing HepG2/HisCore (**b**). The fractionation number (1–14) corresponds to the number of the blot. HBe antigen peaks were determined by ELISA. In all cells, core particles were detected in fractions with a density of around 1.26. An anti-core antibody (bottom panel) detected the core protein in the fraction. (**c**) Purification of HisCore or HisLacZ with Ni-NTA agarose. HisCore and HisLacZ (as a control) were detected in the bound fraction. For the detection of the HisCore protein, an anti-Core antibody was used. For the detection of the HisLacZ protein, an anti-His-tag antibody was used. Input (In), Unbound (Un), Bound (B).

### 3.3. HBV Core Protein Interacts with the MCM2-7 Complex

We noted with interest that the HBV core protein interacted with MCM5 as well as MCM2 in human hepatocytes. The six eukaryotic MCMs usually form a hetero hexamer complex, MCM2-7 [14,15]. To investigate whether the whole MCM2-7 complex is involved in the interaction with the HBV core protein, each Halo™-tagged MCM (HaloMCMs) was individually expressed in the HEK293T cells. The cell lysate of HEK293T/HaloMCM2 was mixed with a lysate of either HepG2/HisCore or HepG2/HisLacZ, and pulled down with Halo™-link resin. As shown in Figure 3, the Halo™-link resin bound HaloMCM2 pulled down all the other endogenous MCMs, including MCM3, 4, 5, 6, and 7 (Figure 3, lanes 4 and 5). In addition, the HaloMCM2 also pulled down HisCore (Figure 3, lane 4), but not HisLacZ (Figure 3, lane 5). Likewise, any of the other MCMs, when Halo-tagged, pulled down HisCore together with the other MCMs (data not shown). These results indicated that any HaloMCMs are functional in order to make a complex with other MCMs, and that the HBV core could interact with one of them.

To further confirm the interaction of the HBV core protein with MCM proteins, the localization of HisCore and HaloMCMs was examined by IFA. As shown in Figure 4, HisCore protein was colocalized with each of the HaloMCMs, mainly in the cytoplasm. It is well known that the MCM2-7 complex localizes in the nucleus during the late G1 to G1/S phase for the initiation of DNA replication; it is, then, exported to the cytoplasm after DNA replication has started to avoid re-replication [16]. These IFA data provided results consistent with the finding that the core protein interacts with MCMs, although they did not show a distinct co-localization due to diffuse distribution.

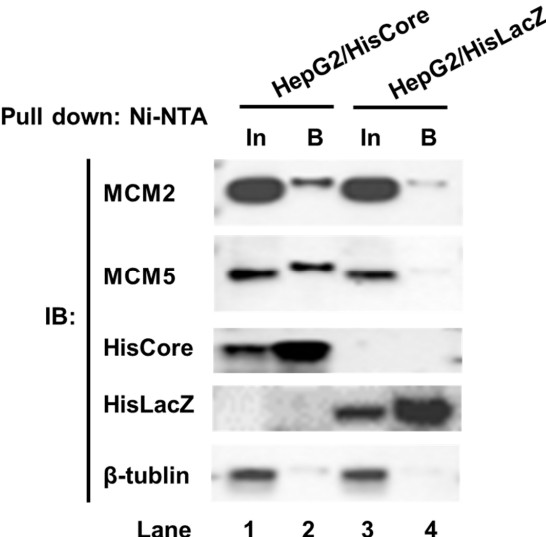

**Figure 2.** Pull-down assay of HisCore with endogenous MCMs. Stably expressing cell lines, HepG2/HisCore and HepG2/HisLacZ cells were used for the pull-down assay. HisCore and HisLacZ proteins were pulled down with Ni-NTA agarose and detected by an anti-His-tag antibody. Endogenous MCM2 and MCM5 were detected by anti-MCM2 antibodies and an anti-MCM5 antibody. Input (In), Bound (B).

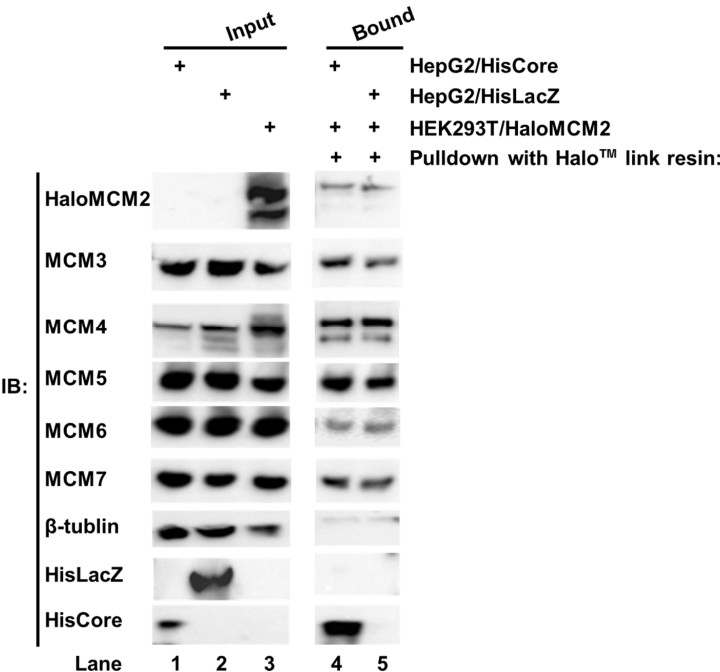

**Figure 3.** Pull-down assay of HisCore with recombinant Halo-MCM2. Halo-tagged MCM2 specifically interacted with the HisCore together with the associated MCMs. Cell lysates of HepG2/HisCore (lane 1), and HepG2/HisLacZ (lane 2) were prepared. The lysate was also prepared from HaloMCM2 transiently expressing HEK293T cells (lane 3). The cell lysate of HEK293T/HaloMCM2 was mixed with the lysate of either HepG2/HisCore or HepG2/HisLacZ and incubated at 4 °C overnight. The bound fraction was washed and eluted with an elution buffer and analyzed by Western blotting. Lane 1, input of HepG2/HisCore; lane 2, input of HepG2/HisLacZ; lane 3, input of HEK293T/HaloMCM2; lane 4, HaloMCM2-bound fraction with HepG2/HisCore; and lane 5, HaloMCM2-bound fraction with HepG2/HisLacZ. "+" indicates the use of those listed to the right. HisCore and HisLacZ proteins were detected by an anti-His-tag antibody. HaloMCM2 was detected by an anti-Halo™-tag antibody. Endogenous MCMs were detected by specific antibodies, as described in Materials and Methods.

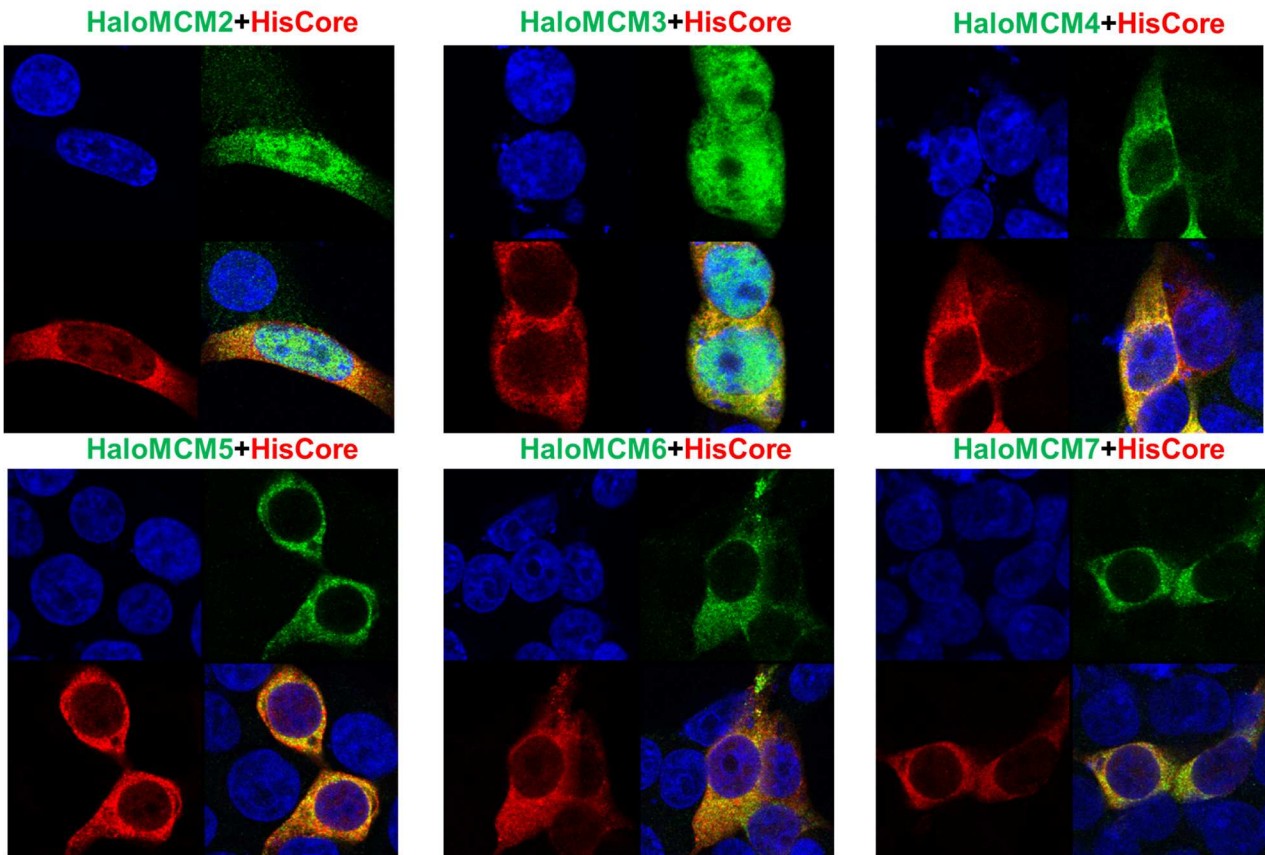

**Figure 4.** Immunofluorescence assay of HisCore and HaloMCMs in HEK293T cells. HisCore and individual HaloMCM were co-transfected into HEK293T cells. HisCore was detected by an anti-Core antibody (Red). HaloMCMs were detected by an anti-Halo™-tag antibody (Green). Nuclei were stained with DAPI (Blue).

### 3.4. Knockdown of MCM2 Leads to Increased Viral Replication

To examine the effect of MCM2 on HBV infection and replication via the interaction with the HBV core protein, we established MCM2-knockdown NTCP/HepG2 cell lines. Knockdown efficiencies were confirmed by Western blot analysis, as shown in Figure 5a. According to the timeline shown in Figure 5b, cells and culture supernatants were collected. Knockdown efficiencies of three established clones were maintained for at least 9 days post-infection (dpi) (Figure 5c). At 9 dpi, HBeAg levels into the supernatants of shMCM2/C4 cells were significantly increased, compared with that of shNC control cells (Figure 5d). Furthermore, levels of cccDNA (Figure 5e), core-associated HBV DNA (Figure 5f), and extracellular particle-associated HBV DNA (Figure 5g) significantly increased, consistent with the level of HBeAg expression. Collectively, these results suggest that MCM2 should have inhibitory effects on HBV proliferation.

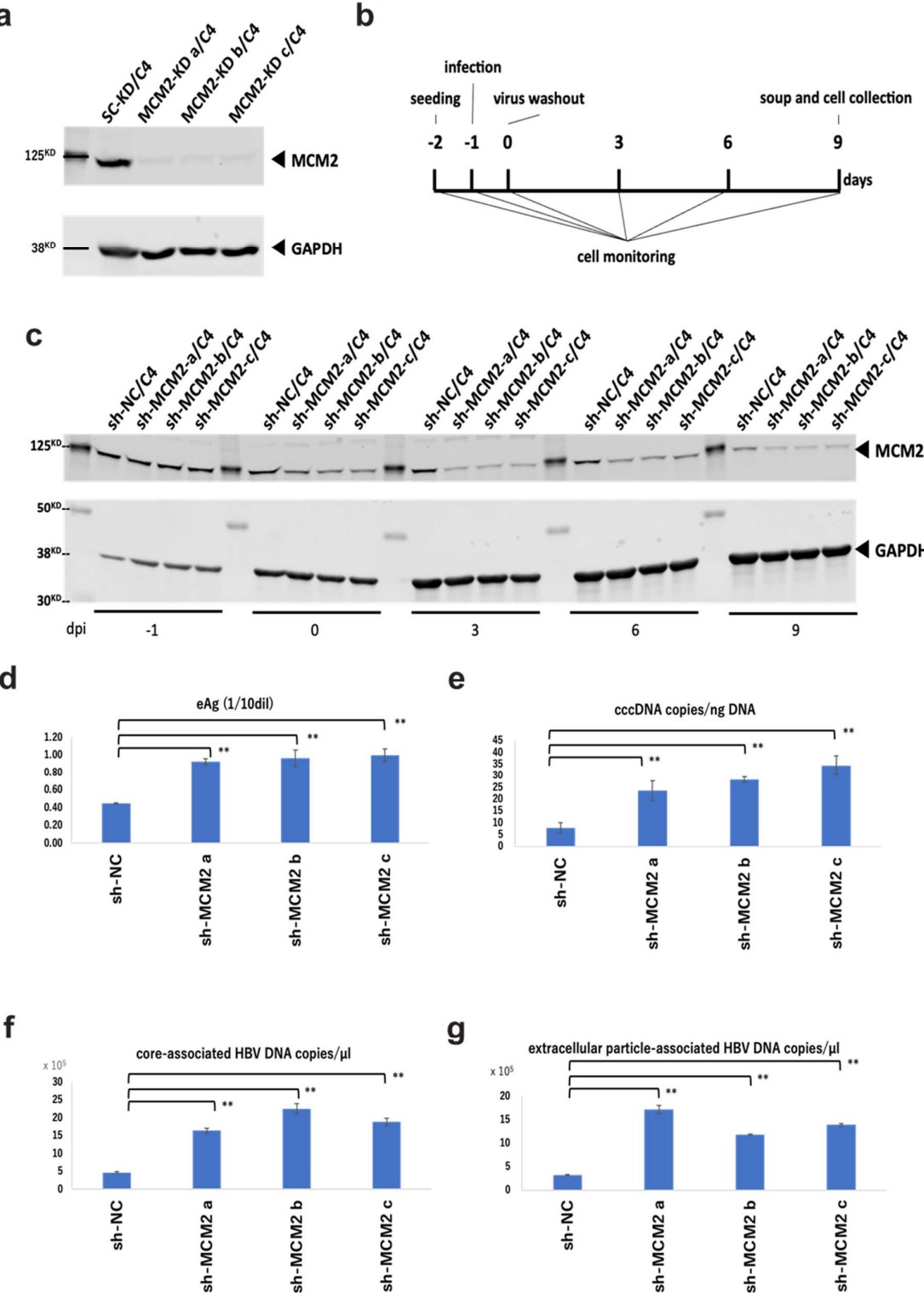

**Figure 5.** Knockdown effects of MCM2 on HBV proliferation during infection. (**a**) Knockdown efficiencies were examined by Western blotting. (**b**) Timeline of the HBV infection in sh-MCM2/C4 cells. (**c**) Knockdown efficiencies were confirmed by Western blotting before and after the HBV infection at −1, 0, 3, 6, and 9 dpi. (**d**) HBeAg levels in the culture supernatants. (**e**) HBV cccDNA copies/ng DNA. (**f**) Evaluation of core-associated HBV DNA (**g**) Evaluation of extracellular particle-associated HBV DNA. Samples in panels (**d**–**f**) and g were collected at 9 dpi. The HBV DNA and cccDNA were quantified by qPCR. Asterisks indicate significant differences between sh-MCM2 and negative control groups (** $p < 0.01$).

## 4. Discussion

It is well known that viruses associate with many host factors to support propagation themselves [17–21], and HBV is no exception. Here, we show that the HBV core protein interacts with the MCM2-7 complex, which has a crucial role in host DNA replication as a DNA helicase [22–26]. This is the first report showing the interaction of HBV core protein with MCMs. On the other hand, many studies have reported that the interaction of MCM complexes or individual MCMs with viral factors affects viral and host replication. The replication of the influenza virus is regulated by the MCM complex [27]. MCM5 is incorporated into HIV-1 virions and regulates HIV replication via association with the Gag polyprotein [28]. The pre-replicative complex (pre-RC), including MCM2-7, is recruited to the Epstein–Barr virus (EBV) replication origin, oriP [24], and Kaposi's sarcoma-associated herpesvirus (KSHV) oriP [29] during latency. KSHV restricts host cellular replication during lytic reactivation by interrupting the MCM complex through ORF59 [30].

In this study, we found that Halo-tagged MCMs were able to specifically pull down the HBV core protein together with the other endogenous MCMs (Figure 3), suggesting that the HBV core protein could interact with MCMs. The eukaryotic MCM complex is evolutionarily conserved and has six components: MCM2-MCM7 [26,31]. These six minichromosome maintenance family proteins form a hetero hexamer complex that is recruited by Cdc6 and Cdt1 to form a stable complex called the pre-replicative complex (pre-RC) for the initiation of DNA replication [23]; in subsequent steps, the DNA is unwound by its helicase activity [14,15,23]. The MCM2-7 proteins are super-abundant in cells, compared with the number of DNA replication origins in chromatin [32–34]. In addition to its helicase activity, it has been reported that MCM2-7 also functions as a component of RNA polymerase II-mediated transcription [34–36] and is responsible for the repair of DNA damage and replication stress via interaction with ATRIP [37,38]. Individual MCM components could be functional independently; for instance, MCM7 interacts with RAD17 for the activation of the DNA damage response (DDR) [39].

IFA suggested that the core protein and MCMs co-localized mainly in the cytoplasm (Figure 4). MCM2-7 is translocated into the nucleus in the late G1 phase, where it forms hexamers and is involved in the initiation of DNA replication as a DNA helicase [40]. When not initiating DNA replication, MCM2-7 is phosphorylated and exported from the nucleus to prevent re-replication [41]. It has been reported that individually and transiently overexpressed MCM4, MCM5, MCM6, and MCM7 are mainly localized in the cytoplasm in asynchronous cells, while MCM2 and MCM3 are distributed inside the nucleus because of their distinct nuclear localization signals (NLSs) [40]. Consistent with this study, our IFA results showed that MCM2 and MCM3 localized in both the cytoplasm and nucleus (Figure 4).

It has been recently reported that the re-phosphorylation of core proteins at the N-terminal domain (NTD) and the C-terminal domain (CTD) by cyclin-dependent kinase 2 (CDK2) facilitates the disassembly of nucleocapsids and cccDNA formation [42]. It was reported that CDK2 directly interacts with several core factors of the MCM complex, including MCM2, MCM6, and MCM7, and plays a role in the appropriate DNA replication [43,44]. From our results, MCM2 knockdown significantly increased the levels of HBeAg, cccDNA, core-associated HBV DNA, and extracellular particle-associated HBV DNA during the HBV infection. The results suggested that MCM2 should be involved in HBV entry/establishing an infection or cccDNA accumulation in the internal cycle [45–48], and that the MCM2 should serve as a restriction factor for HBV replication (Figure 5).

The HBV core protein carries a nuclear localization signal and is a component of cccDNA minichromosomes [49,50]; however, very little is known about the function of the HBV core protein inside the nucleus. Although it might be a minor population of core proteins, the possibility that the interaction between core proteins and MCMs occur in the nucleus should not be denied. Further analysis is required to discuss the functional mechanisms of the interaction between the HBV core protein and MCM2-7; however, it might interact with some of these diverse functions of MCM2-7, and these interactions

might be involved in parts of the HBV lifecycle. Since the HBV core protein induces the production of host interferon and cytokines, which have been implicated in chronic HBV infection and hepatocarcinogenesis, HBV core inhibitors, such as capsid assembly inhibitors, could contribute to the prevention of liver disease progression in HBV-infected patients [51]. The HBV core protein is an attractive target for the treatment of chronic hepatitis B because of its multiple roles in HBV replication; therefore, further elucidation of its function in viral replication, through the analysis of interacting factors, will be helpful in the development of new therapeutic strategies.

**Author Contributions:** Conceptualization, K.D. and K.U.; methodology, K.D., E.O. and M.W.; validation, K.D.; formal analysis, K.D.; investigation, K.D., E.O. and M.W.; resources, K.U.; data curation, K.D.; writing—original draft preparation, K.D.; writing—review and editing, E.O. and K.U.; supervision, K.U.; project administration, K.U.; funding acquisition, K.U. All authors have read and agreed to the published version of the manuscript.

**Funding:** K.D. was supported by the Interdisciplinary Program for Biomedical Sciences, Program for Leading Graduate School, Osaka University Grants-in-Aid for Education and Research, Japan. This work was supported by grants from the Japan Agency for Medical Research and Development (AMED) to K.U. (17fk0310105h0001, 18fk0310105h0002, 19fk0310105h0003, 20fk0310105h0004, 21fk0310105h0005, 22fk0310505h0001).

**Institutional Review Board Statement:** Not applicable.

**Informed Consent Statement:** Not applicable.

**Data Availability Statement:** Not applicable.

**Acknowledgments:** The authors gratefully acknowledge H. Otake for her invaluable technical assistance with the experiments.

**Conflicts of Interest:** The authors declare no conflict of interest.

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
