# Peer review of "Identification of the Interaction between Minichromosome Maintenance Proteins and the Core Protein of Hepatitis B Virus"

_cimb, doi:10.3390/cimb45010050_

Round 1

Reviewer 1 Report

Du et al. reported that the possibility of elucidating the relationship between core proteins and host factors and their function in viral proliferation.

How the effects of HBV core protein on host immune reaction? How the effects of HBeAg on them? Authors should discuss more. See the reference: Kanda T, et al. HBV Core Protein Enhances Cytokine Production. Diseases. 2015 Sep 17;3(3):213-220. doi: 10.3390/diseases3030213. PMID: 28943621

Reviewer 2 Report

This manuscript reported the interaction of the core protein with MCMs. Moreover, the Knockdown of MCM2 led to increased viral replication, suggesting that MCM2 serves as a restriction factor for HBV proliferation.

 Several suggestions:

1.      References are not correct, such as 1,2, 3, etc. Please check all the references to make sure that they are properly cited.

2.      Fig. 1a and 1b, why there is HBeAg? Did HepG2/HisCore cells express pre-core?

3.      Fig. 2, why the size of MCM2 is different between lanes 1 and 2? The size of MCM5 is also different between lanes 1 and 2. This figure is not convincing because both MCM2 and MCM5 in lane 4 are so visible.

4.      Figure 4. It is better to detect the co-localization of endogenous MCM2 and core protein in HepG2/HisCore cells. Is it difficult to conduct this experiment?

5.      Fig. 5, Knockdown of MCM2 led to increased HBV viral replication. How about over-express MCM2?

Round 2

Reviewer 2 Report

The authors have addressed the issues I have raised previously in this revised manuscript.